# Investigating stronger tolerant network against cascading failures in focusing on changing degree distributions

**Ryota Kusunoki** ☉*, **Yukio Hayashi** ☉

Graduate School of Advanced Science and Technology, Japan Advanced Institute of Science and Technology, Nomi, Ishikawa, Japan

☉ These authors contributed equally to this work.

* ryotakusunoki@hotmail.com

## Abstract

Many real-world networks with Scale-Free structure are significantly vulnerable against both intentional attacks and catastrophic cascading failures. On the other hand, it has been shown that networks with narrower degree distributions have strong robustness of connectivity by enhancing loops. This paper numerically reveals that such networks are also tolerant against cascading failures. Our findings will be useful in designing stronger tolerant network infrastructures.

**Data Availability Statement:** The minimal data set and code can be accessed at https://github.com/sonokr/cascading_failures_simulation.

**Funding:** YH is partially supported by a Grant-in-Aid for Scientific Research (Grant Number

## Introduction

In the early 20th century, Erdős–Rényi (ER) random graphs with homogeneous Poisson degree distributions were widely studied with mathematical interests [1, 2]. After decades, through the progress of techniques for large data analysis, it has been found that Scale-Free (SF) structure with significantly heterogeneous power-low degree distribution exists in many real-world networks such as electric power-grid, actor collaborations, neuronal network [3, 4]. Unfortunately, SF networks are extremely vulnerable against intentional attacks to large degree nodes [5]. On the other hand, by the analysis of generating function and numerical simulations, it has been also revealed that, under a given degree distribution, onion-like structure has the optimal robustness of connectivity when degree-degree correlations increase moderately [6, 7]. However, the robustness of connectivity is decreased for too large positive correlations. Thus, degree-degree correlations may be not essential for improving the robustness.

Recently, it has been suggested that loops are more important than degree-degree correlations for the robustness of connectivity against attacks [8, 9]. Bacause the importance of loops is theoretically supported by the asymptotic equivalence of the dismantling and recycling problems [9], when the second moment of degree is not divergent. Roughly speaking, the worst case attacks for fragmentation is to be loopless. Conversely, the robustness becomes higher as the size of necessary nodes to form loops is larger, although even only measuring of the size is NP-hard in computer science. Actually, by enhancing loops, several rewirings

JP.21H03425) from the Japan Society for the Promotion of Science. The funders had no role in study design, data collection and analysis, decision to publish, or preparation of the manuscript. There was no additional external funding received for this study.

**Competing interests:** The authors have declared that no competing interests exist.

without preserving degrees have been proposed to improve the robustness of connectivity [10]. In particular, even for different rewiring methods, the decreasing the gap between the maximum and the minimum degrees has been found commonly. Thus, narrower degree distribution leads to increase the robustness of connectivity.

As a related study to change degree distributions, growing network (GN) model [11] generates networks with continuously changing degree distributions between power-law and exponential ones, although the robustness of connectivity is not discussed. Moreover, inverse preferential attachment (IPA) model has been recently introduced [12], and generates networks with narrower degree distribution than exponential one. Instead of the gap between the maximum and the minimum degrees, the narrowness can be measured and compared by the variance $\sigma^2 = \Sigma_k (k - <k>)^2 P(k)$ for any degree distribution P(k) with average degree $<k> = \Sigma_k k P(k)$. In particular, it has shown that the robustness of connectivity against attacks increases more as narrower degree distribution [10]. It is also suggested that, the extreme case of the narrowest degree distribution whose width is zero, random regular graphs are the optimal for the robustness [13]. Note that randomization is necessary to investigate the pure effect of the degree distribution, since the IPA model has a chain-like structure [12]. As examples in real networks, exponential and narrower degree distributions appears in growing social networks in which new-comers connect to randomly chosen nodes with encountering and road networks with T-junctions and cross-roads, respectively.

On the other hand, even if a network is connected, the malfunction as overload may be occurred through redistribution of flows triggered by the removal of one or a few nodes. Where the flows are corresponded to moving vehicles on road or transferring packets in communication, and the overload means that the node exceeds its capacity. This is called cascading failures and represents a phenomenon of chain reaction to widespread malfunctions in many network systems. Even when links remain intactly, some load exceeded from the capacity for processing or flow passage at nodes results in the loss of connectivity potentially causing severe damage. Excluding their minor differences, this mechanism becomes common for such as power grid collapses, communication congestions, and cascading bankruptcies in infrastructures that widely sustain our contemporary society. Therefore, the problem of cascading failures is very serious. So far cascading failures and defense strategies have been discussed for classical ER random graphs and realistic SF networks, however they are pin-point researches for only Poisson and power-low degree distributions. Note that SF network is significantly vulnerable against cascading failures triggered by intentional attacks [14], and that ER random graph is relatively tolerant compared to SF network [15]. While several defense strategies have been proposed as sacrificial node removal [16], reconnection of links [17], and improving of routing based on degree and load [18], these defense strategies are impractical. Because nodes and links are wasted on the assumption that the network structure can be changed immediately. Furthermore, in recent years, cascading failures have been extendedly studied on interdependent networks, whose infrastructure consists of power-grid, communication, and social networks [19–21]. However, even on a single layered network, the stronger tolerant structure is unknown against cascading failures.

Therefore, away from the pin-points of the Poison and power-low degree distributions, we investigate more tolerant topological structures against cascading failures. We focus on changing degree distribution, since a narrower degree distribution leads to enhancing many loops. Therefore, the tolerance against cascading failures is expected because of distributed flows via many bypass routes on loops. Through numerical simulations, we reveal that networks with narrower degree distributions are also higher tolerant against cascading failures.

## Continuously changing degree distributions

We consider preferential attachment (PA) [3] and IPA [12] in growing networks. It is well-known that Barabási-Albert (BA) model [3] generates a heterogeneous SF network by PA, whose attachment probability is proportional to degree $k_i$ of existing node $i$. Here, the attachments make $m$ links from a new node to existing nodes in a network at each time step. While in IPA model, the probability is proportional to $k^{-\beta}, \beta \geq 0$ [12], the generation process is summarized as follows.

**Step 1**. Start with a complete graph of $2m = 4$ nodes as initial configuration.

**Step 2**. At each time step, a new node is added and attached to existing nodes with $m = 2$ links.

**Step 3**. Repeat Steps 1, and 2 until the the network size reaches $N$.

By IPA model as larger $\beta$, network efficiency described in the next subsection is significantly decreased, because a special chain-like structure emerges [12, 22]. Therefore, configuration model [23] is applied as randomization in order to eliminate such a chain-like structure and to investigate the pure effect of degree distributions.

Fig 1 shows the degree distributions in heterogenous SF (dashed black line), homogeneous ER random graphs (dash-dot black line), and more homogeneous (colored lines) networks. The corresponding degree distributions are power-low, Poisson, and narrower ones, respectively. Note that the dashed black line for SF network is straight in log-log plot. As $\beta$ increases, the maximum degree generated by IPA is decreasing. In particular, networks for $\beta = 50$ (orange), and 100 (red) are very homogeneous with narrower degree distributions. The Poisson distribution [1] in ER random graph (dash-dot black line) is similar to that for the case of $\beta = 1$ in IPA (light blue line). Note that the case of $\beta = 0$ by IPA (blue) is an exponential degree distribution generated by random attachment.

In general, cascading failures can be suppressed with many bypass routes in the network. In contrast, the existence of bypass routes is measured by the distribution of link betweenness centrality (BC) [24]. If there are the shortest paths between any nodes as bypass routes, the width of BC distribution becomes narrow. Here, the value of link BC is defined by the ratio of

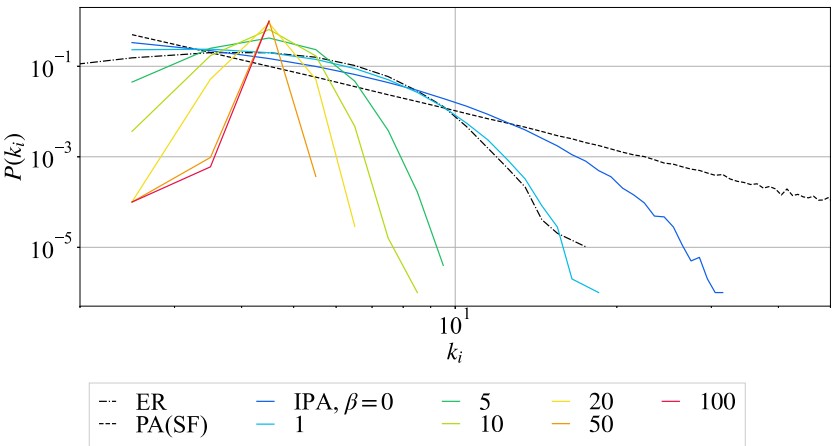

**Fig 1. Degree distributions for the network size $N = 10^3$ and the total number of links $M = 2000$.** The dash-doted black line represents a Poisson degree distribution for the network generated by ER random graph, and the dashed black line represents a power-low degree distribution for the network generated by PA. Colored lines represent the narrower degree distributions for networks generated by IPA with different value of $\beta$ denoted from blue to red.

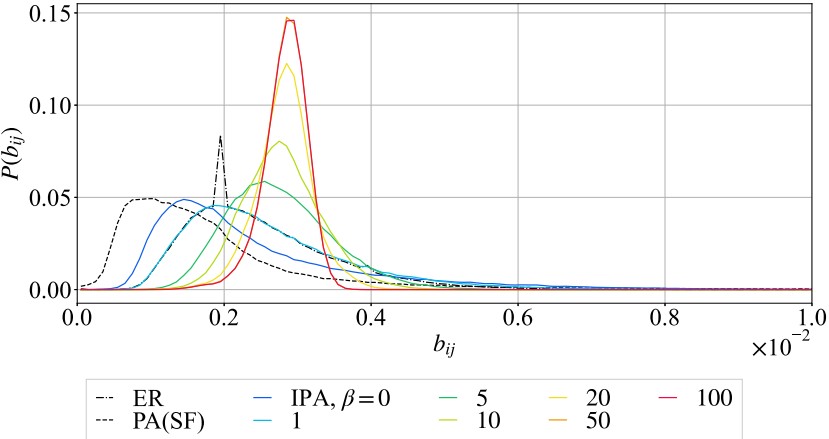

**Fig 2. Distributions of link betweenness centralities.** The other parameters and the networks represented by lines are the same as in Fig 1.

the number of shortest paths through link $(i, j)$ to the total number of shortest paths between source and destination nodes $s, d \neq i, j$. Fig 2 shows the distributions of link BC in heterogeneous SF networks generated by PA (BA model) (dashed black line), homogeneous ER random graphs (dash-dot black line), and more homogeneous networks generated by IPA (colored lines). As changing from SF by PA (dashed black line) to the case of $\beta = 100$ by IPA (red line), many links tend to have similar values of link BC with the smaller variance of link BC distribution. In other words, it is suggested that more bypass routes exist in networks with the narrower degree distribution generated by IPA.

It has been shown that, in a modification of IPA model, small amount of random attachment is necessary for the emergence of the average path length $\langle L_{ij} \rangle \sim O(\log N)$ as Small-World (SW) property [25] to eliminate chain-like structure for large $\beta$. Instead of random attachment, we apply randomization by configuration model [23] to investigate a pure effect of degree distribution. Fig 3 shows the average shortest path length in randomized networks without chain-like structure [12]. In the left of Fig 3, all lines are straights of $O(\log N)$ in a semi-log plot as the SW property [25]. However, without the randomization in the right side of Fig 3, the shortest path length increases exponentially as $\beta$ increases because of the emergence of

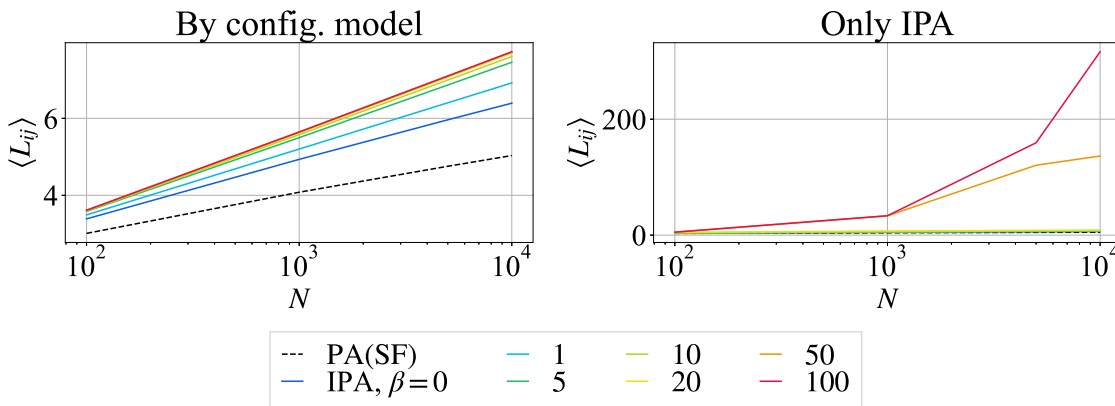

**Fig 3. Average shortest path lengths $\langle L_{ij} \rangle$ for the network size $N = 10^4$.** (Left) randomized by configuration model, (Right) networks generated by only IPA. The other parameters and the network represented by lines are the same as in Fig 1.

chain-like structure [12]. Thus, it is expected that networks with narrower degree distributions have both high tolerance and network efficiency against cascading failures as shown later, since the shortest path length is comparable to that in SF networks. The definition of tolerance and network efficiency against cascading failures are explained in the next section.

## Cascading failures

We explain a typical procedure of cascading failures [14]. The load $L_i(t)$ is defined by node betweenness [26] on the assumption that flows are transmitted along the shortest paths between source $s$ and destination $d$ nodes at where the unit quantity of transmission request is generated in each pair of nodes.

$$L_i(t) = \sum_{s \neq d \neq i} \frac{g_{sd}(i)}{g_{sd}}, \tag{1}$$

where $g_{sd}$ is the total number of shortest paths from $s$ and $d$ nodes, $g_{sd}(i)$ is the number of those paths passing through node $i$ at time step $t \geq 0$. Node betweenness means the load as the ratio of flows passing through node $i$ at $t$ and is effectively calculated by Brandes algorithm [27]. Eq (1) implies that the flow is equally distributed to multiple shortest paths with a same length. We remark that link BC is not used to define the load at a node but applied to measure the amount of bypass routes. At each time step $t \geq 1$, $g_{sd}(i)$ and $g_{sd}$ are changeable after node removals.

The capacity $C_i$ is defined as proportional to the initial load $L_i(0)$.

$$C_i = (1 + \alpha)L_i(0), \tag{2}$$

where $\alpha > 0$ is a tolerance parameter, and $L_i(0)$ denotes the initial load at node $i$ in the network at time step $t = 0$. When $L_i > C_i$: the load $L_i$ exceeds the capacity, node $i$ is removed from the network as a malfunction. After such node removals, Eq (1) is recalculated for $t \to t + 1$ from $t = 0$. In other words, the load is redistributed to the remaining nodes after re-routing on the shortest paths. Then, nodes with $L_j > C_j$ are further removed from the network. Such process is repeated until $L_k < C_k$ is satisfied for remaining all nodes.

We consider the following three typical methods [14] and spatial damage methods [28] as triggers of cascading failures, since the damage of cascading failures depends on the types of triggers.

- Intentional attacks: Max-degree, Max-load attacks [14]

- Unexpected failures: Random attacks [14]

- Spatial damages: Localized attacks [28]

Max-degree and max-load attacks choose the nodes with the maximum degrees and loads in all nodes, while random attacks choose nodes uniformly at random as unexpected failures, respectively. As spatial damage by earthquakes or tsunamis [28], localized attacks remove connected nodes based on their hop-distances from a root node. The root node is chosen by three types of triggers: random, max-degree, and max-load attacks, respectively. We set the number of initially removed nodes as $AN = N \times p$, where $p$ is the rate of node removals.

The damage of cascading failures is quantified by the relative size $G$ of the largest connected component (LCC) before and after cascading failures.

$$G = \frac{N'}{N}, \tag{3}$$

where $N'$ is the the network size in the LCC after cascading failures. Furthermore, network efficiency is introduced to measure the efficiency of paths [29]. The average efficiency is defined as,

$$E = \frac{1}{N(N-1)} \sum_{i \neq j} e_{ij}, \tag{4}$$

where the efficiency $e_{ij}$ between nodes $i$ and $j$ is inversely proportional to the distance $d_{ij}$ of the shortest path length $e_{ij} = 1/d_{ij}$. For example, $d_{ij}$ is calculated by Dijkstra [30] or the preprocessing of Brandes algorithm [27]. If there is no path between the nodes $i$ and $j$, we set $d_{ij} = \infty$ and $e_{ij} = 0$. Note that the efficiency is defined as the inverse of the harmonic mean, while the average path length is as the arithmetic mean.

We summarise the above process of cascading failures.

1. Calculate the load $L_i$ on each node $i$ by Eq (1).

2. Remove trigger nodes chosen by random, Max-degree, or Max-load node removals.

3. Recalculate the load $L_i$ on each node $i$ after re-routing.

4. When $L_i > C_i$, the nodes whose load exceeds their capacity are removed from the network.

5. Repeat Steps 3, 4 until all nodes are removed or there are no overloaded nodes.

Remember that, after removing either AN = 1 or 50 nodes as triggers, the flows between $s$ and $t$ are equally distributed to the multiple shortest paths with a same length by Eq (1) in the determinant choosing. Therefore, stochastic choosing from the multiple shortest paths is beyond our scope of this study.

## Results

As a degree distribution becomes narrow, the network becomes more robust in enhancing loops [10]. Thus, it is expected that many loops potentially lead to increased the tolerance against cascading failures because of distributed flows via many bypass routes on multiple loops. In this section, we will provide a numerical validation of the tolerance against cascade failures. Cascading failures are simulated for the heterogeneous SF networks generated by PA, homogeneous ER random graphs, and the more homogeneous networks generated by IPA for finding more tolerant network structure in changing degree distributions. Remember that the corresponding degree distributions are power-low, Poison, and narrower ones. In order to investigate the pure effect of the degree distributions, networks generated by IPA are rewired at random by configuration model. We set the network size $N = 1000$, the total number of links M = 2000, $\beta = 0, 1, 5, 10, 20, 50, 100$ in IPA, tolerance parameter $\alpha = 0, 0.1, 0.2, 0.3, 0.4, 0.5, 0.6, 0.7, 0.8, 0.9, 1.0$, and number of removal nodes $AN = 1, 50,$ or $100$. The following results are averaged over 100 realizations of randomly generated networks. Note that the results for max-degree and max-load attacks are similar, because the node with the max-degree tends to have the max-load. These similar results are not shown but are presented in supporting information.

Fig 4 shows the relative sizes $G$ after cascading failures. The dash-dotted black line, dashed black line, and colored lines represent the relative sizes $G$ for ER random graph, SF network, and the more homogeneous networks randomized by configuration model after IPA in order to investigate the pure effects of degree distributions. The colored lines represent the relative sizes $G$ for network by configuration model after IPA for different values of $\beta$ denoted from blue to red in continuously changing degree distributions.

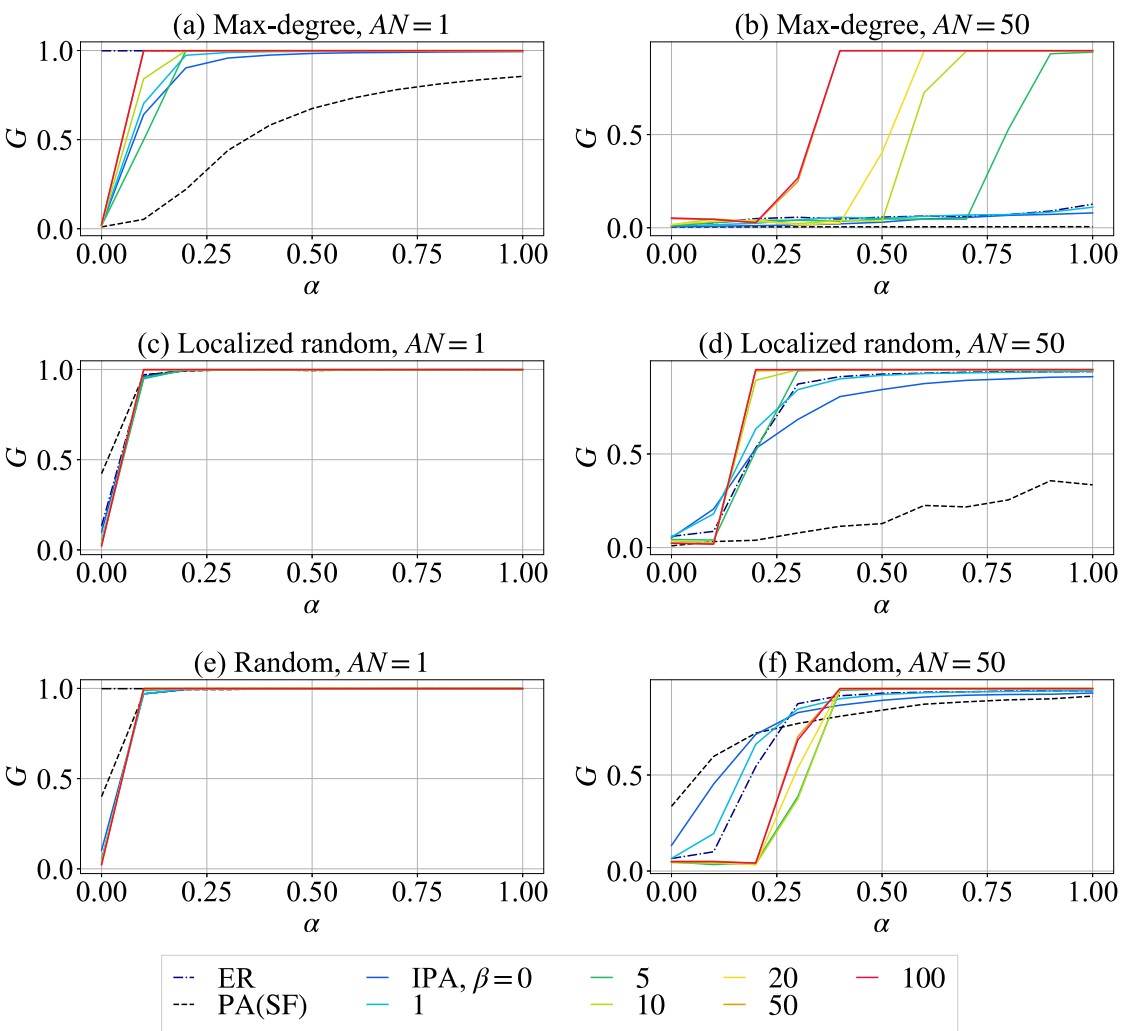

**Fig 4. Relative size *G* for networks with *N* = 1000 and *m* = 2 by configuration model.** (Top) random attacks, (Center) max-degree attacks, (Bottom) localized random attacks. (Left) number of initial node removal is *AN* = 1, (Right) number of initial node removals is *AN* = 50.

In Fig 4(a) the relative size *G* of the network (colored lines) randomized by configuration model after IPA is more highly tolerant than that of SF network (dashed black line) against max-degree attacks for both *AN* = 1 (a) and 50 (b). The case of larger *β* from blue to green, yellow, and red suppresses the damage of cascading failures for *AN* = 50 (b) even with a small value of tolerance parameter. This indicates that the tolerance against cascading failures increases by distributing the load among nodes on the multiple shortest paths, because of many loops between any nodes as the degree distribution becomes narrower.

Against the localized random attacks as shown in Fig 4(d), SF network (dashed black line) has a smaller relative size *G* than that against random attacks as shown in Fig 4(f) for *AN* = 50. However, as larger *β*, the relative size *G* is larger in the networks (colored lines) by configuration model after IPA. SF network is vulnerable with a high probability that attacked nodes include hubs due to the short path lengths and heterogeneous degree distribution. On the other hand, path lengths between any nodes are short but no hubs exist, if the degree

distribution is sufficiently narrow. Therefore, the networks have a higher tolerance against cascading failures. Note that only the root node is removed by localized random attacks for $AN = 1$, the relative size $G$ is the same as that by random attacks.

On the other hand, in Fig 4(e), the relative size $G$ is almost 1.0, when the number of initial random attacks is $AN = 1$. It means that there is no major damage against cascading failures except for the case of small $\alpha < 0.1$. While for $AN = 50$ in Fig 4(f), SF network (dashed black line) is more tolerant than other networks against cascading failures, as similar to the high robustness of connectivity from the SF network against random attacks [5]. In other words, it is considered that the damage of cascade failures are suppressed as maintaining of connectivity after initial attacks. Furthermore, the relative size $G$ of networks randomized by configuration model after IPA is larger for $\alpha \geqq 0.4$ than that of SFs network. Note that the relative size $G$ of ER random graph (dash-dot black line) is close to that of the network by configuration model after IPA with $\beta = 1$ (light blue line).

Fig 5 shows the network efficiency after a cascading failure by random and localized random attacks for $AN = 50$. On the right of Fig 5, SF network (dashed black line) is significantly inefficient against localized random attacks, while it is the most efficient against random attacks. However, against random attacks, the efficiency of the SF network is slightly higher than the network randomized after generated by IPA. In these networks with long-tailed and narrower degree distributions, the difference of network efficiency is at most approximately 0.05 on the left of Fig 5. Therefore, more homogeneous networks are not only higher tolerant against cascading failures, but also become efficient for path lengths.

Fig 6 illustrates the network efficiency $E$ before and after cascading failures triggered by localized random, max-degree, and max-load node removals in randomized networks after generations by PA and IPA in changing degree distribution. Before cascading failures (blue line), SF network is the most efficient, while networks by IPA converge to approximately $E = 0.195$ around $\beta = 10$. However, after cascading failures triggered by localized-random node removal (magenta line), the efficiency of SF networks significantly decreases. In contrast, networks by IPA show only a slight reduction in efficiency across any $\beta$ values. Furthermore, after cascading failures triggered by max-degree and max-load removals (green and dashed red lines), the efficiency shows a remarkable similarity of maintaining a consistent level at approximately $E = 0.190$ when $\beta \geq 5$. Overall, the homogeneous networks generated by IPA present a decrease in efficiency at most approximately 30%, compared to SF networks before cascading failures which is the most efficient. In addition, at $\beta = 100$ when the network

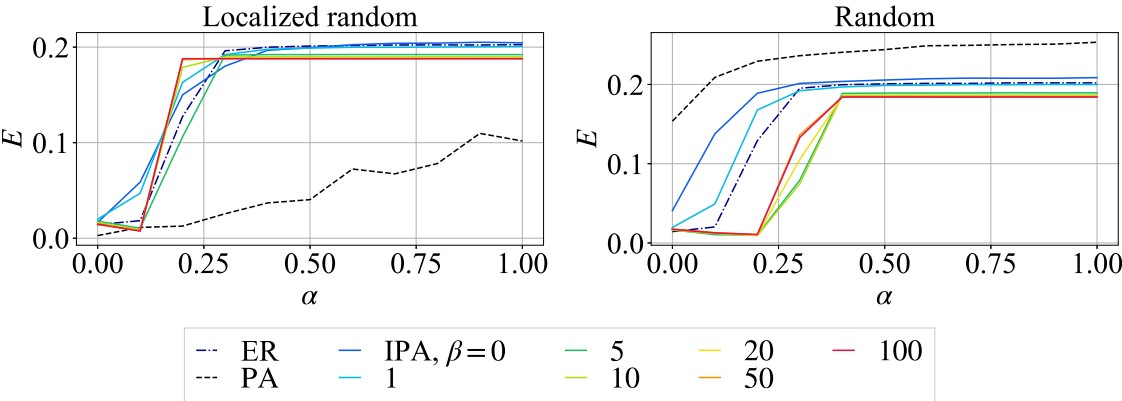

**Fig 5. Network efficiency $E$ in networks for $N = 1000$, $m = 2$, and $AN = 50$ by configuration model.** (Left) Random attacks, (Right) Localized random attacks.

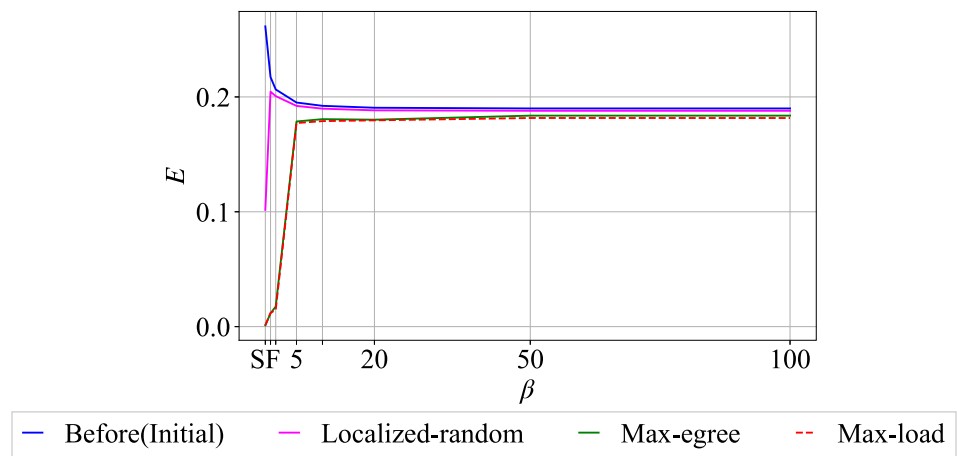

**Fig 6. Network efficiency _E_ before and after cascading failures by localized random, max-degree and max-load nodes removal for _N_ = 1000, _m_ = 2, and _AN_ = 50.**

achieves sufficient homogeneity, the difference of efficiency between before and after cascading failures is very small.

## Conclusion

In this study, we explored the relation between network topology characterized by degree distribution and the tolerance against cascading failures. We consider the cascading failures from various triggers, such as intentional attacks, random failures, and spatial damages. Our results show that narrower degree distributions absorb their cascading failures. Specifically:

1. The tolerance against cascading failures is significantly improved as the degree distribution narrows. Because flows are distributed to bypass routes in the networks, it mitigates the risk of cascading failures effectively.

2. In contrast, networks with narrower degree distributions not only have in tolerance cascading failures but also manifested efficiency for their path lengths. SF network is most efficient before cascading failures, but the efficiency is reduced at most approximately 30%. It is also worth to note that the difference between before and after cascading failures by any node removals is negligible for sufficiently homogeneous networks by IPA for $\beta > 10$.

3. While SF network characterized by a power-law degree distribution has tolerance against random attacks, it displayed vulnerability against max-degree and localized random attacks due to the concentration of load at hubs.

This study revealed a tolerant network structure against cascading failures. Such insights will be useful in designing more tolerant infrastructure against natural disasters or damages by intentional conflicts. The reduction of efficiency after cascading failures is the minimum, therefore the networks can be expected to recover rapidly even in cases of malfunction. Further discussions for vulnerabilities and strengths against cascading failures are remaining in other networks with different degree distributions from ours generated by PA or IPA models. Notably, a recent perturbation analysis suggested that the most robust network structure is exhibited by random regular graphs with 0 variance of degree distribubion [13]. Exploring the relationship between efficiency and other network characteristics, such as the clustering coefficient, remains future works.

## Supporting information

**S1 Fig. Relative size *G* and network efficiency *E* against random attacks in randomized networks for *N* = 1000 and *m* = 2 by configuration model.**
(EPS)

**S2 Fig. Relative size *G* and network efficiency *E* against max-degree degree attacks in randomized networks for *N* = 1000 and *m* = 2 by configuration model.**
(EPS)

**S3 Fig. Relative size *G* and network efficiency *E* against max-load degree attacks in randomized networks for *N* = 1000 and *m* = 2 by configuration model.**
(EPS)

**S4 Fig. Relative size *G* and network efficiency *E* against localized random attacks in randomized networks for *N* = 1000 and *m* = 2 by configuration model.**
(EPS)

**S5 Fig. Relative size *G* and network efficiency *E* against localized max-degree degree attacks in randomized networks for *N* = 1000 and *m* = 2 by configuration model.**
(EPS)

**S6 Fig. Relative size *G* and network efficiency *E* against localized max-load degree attacks in randomized networks for *N* = 1000 and *m* = 2 by configuration model.**
(EPS)

## Author Contributions

**Conceptualization:** Yukio Hayashi.

**Funding acquisition:** Yukio Hayashi.

**Investigation:** Ryota Kusunoki, Yukio Hayashi.

**Methodology:** Ryota Kusunoki, Yukio Hayashi.

**Supervision:** Yukio Hayashi.

**Visualization:** Ryota Kusunoki.

**Writing – original draft:** Ryota Kusunoki.

**Writing – review & editing:** Ryota Kusunoki, Yukio Hayashi.

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
