## [Decision Letter · Decision Letter 0]

12 Jun 2023

PONE-D-23-15565Investigating stronger tolerant network against cascading failures in focusing on changing degree distributionsPLOS ONE

Dear Dr. Kusunoki,

Thank you for submitting your manuscript to PLOS ONE. After careful consideration, we feel that it has merit but does not fully meet PLOS ONE’s publication criteria as it currently stands. Therefore, we invite you to submit a revised version of the manuscript that addresses the points raised during the review process.

We look forward to receiving your revised manuscript.

Kind regards,

Qing-Chang Lu

Academic Editor

PLOS ONE

Journal Requirements:

"This research is supported in part by JSPS KAKENHI Grant Number JP.21H03425."

"YH is partially supported by a Grant-in- Aid for Scientific Research (Grant Number JP.21H03425) from the Japan Society for the Promotion of Science. The funders had no role in study design, data collection and analysis, decision to publish, or preparation of the manuscript. There was no additional external funding received for this study."

3. Please upload a copy of Figures 1,4 and 5, to which you refer in your text on pages 3,6 and 7. If the figure is no longer to be included as part of the submission please remove all reference to it within the text.

Additional Editor Comments:

The reviewers recommend reconsideration of your manuscript following major revisions. I invite you to resubmit your manuscript after addressing the reviewers' comments.

When revising your manuscript, please consider all issues mentioned in the reviewers' comments carefully: please outline every change made in response to their comments and provide suitable rebuttals for any comments not addressed.

Reviewers' comments:

Reviewer's Responses to Questions

**Comments to the Author**

1. Is the manuscript technically sound, and do the data support the conclusions?

Reviewer #1: Partly

Reviewer #2: Yes

2. Has the statistical analysis been performed appropriately and rigorously? 

Reviewer #1: Yes

Reviewer #2: Yes

3. Have the authors made all data underlying the findings in their manuscript fully available?

Reviewer #1: No

Reviewer #2: Yes

4. Is the manuscript presented in an intelligible fashion and written in standard English?

Reviewer #1: Yes

Reviewer #2: Yes

5. Review Comments to the Author

Reviewer #1: This study focuses on the resilient structure to mitigate cascading failures, which is an interesting topic. However, the main concern about this study is the lack of creativity.

The main concern about this study is that the changing degree distributions of network influence the cascading failures are not modeled in the cascading failures model?

The results at the end of the Abstract should be specified. The result that “networks with narrower degree distributions are stronger tolerant against cascading failures.” which is similar with the statement at the beginning of Abstract that “homogeneous networks with narrower degree distributions have many loops and strong robustness of connectivity”

What does the result "In addition, the networks have high efficiency similar to SF networks" in the Abstract specifically mean?

There are several abbreviations that require explanation, for example, BA.

Reviewer #2: I have carefully read your paper and I have some comments and suggestions for improvement.

First of all, I appreciate your original contribution to the field of network science and your interesting approach to enhance the robustness of scale-free networks by changing their degree distributions. Your numerical simulations show that networks with narrower degree distributions have higher tolerance against cascading failures and higher efficiency than scale-free networks.

However, I also have some concerns and questions about your paper that need to be addressed before it can be accepted for publication. Here are some of them:

1.The abstract is too long and contains too many details that should be moved to the introduction or the methods section. Please shorten the abstract to about 150 words and focus on the main motivation, results and implications of your work.

2.The introduction lacks a clear statement of the research problem and the research gap that your paper aims to fill. Please provide more background information on the existing literature on network robustness and cascading failures, and explain how your work differs from or extends previous studies.

3.The methods section is not well explained and lacks some important information. For example, how did you generate the networks with different degree distributions? What are the parameters and assumptions of your cascading failure model? How did you measure the efficiency of the networks? Please provide more details and references for your methods and justify your choices.

4.The results section is too descriptive and does not provide enough analysis and interpretation of your findings. For example, why do networks with narrower degree distributions have higher tolerance and efficiency than scale-free networks? What are the underlying mechanisms or principles that explain this phenomenon? How do your results compare with other studies on network robustness and cascading failures? Please discuss these aspects more thoroughly and provide some figures or tables to illustrate your results.

5.Cascading failures is a complex process, and under the same attack, the evolution of cascading failures can have multiple possible paths. How does your research take this into account? And please elaborate on the multiple evolution processes of network cascading failures under three different attacks.

6.In my opinion, for the resilience of complex networks, it is not that the narrower the degree distribution, the better, nor is it the wider the better. Instead, it is about finding its balance point. Your article does not explain this part very clearly. How narrow should the degree distribution be for optimal resilience?

6. PLOS authors have the option to publish the peer review history of their article (what does this mean?). If published, this will include your full peer review and any attached files.

Reviewer #1: No

Reviewer #2: No

---

## [Author Response · Author response to Decision Letter 0]

30 Sep 2023

I hope this message finds you well. Thank you for taking time to review our manuscript titled "Investigating stronger tolerant network against cascading failures in focusing on changing degree distributions" submitted to PLOS ONE under Manuscript ID PONE-D-23-15565. We greatly appreciate your insightful feedbacks, which have significantly contributed to improving the quality of our work.

We have carefully considered each of your comments and suggestions, and we are pleased to submit the revised version of our manuscript along with this response letter outlining the changes we have made in response to your feedback.

We explain the revisions made:

For Reviewer #1

Q1. The main concern about this study is that the changing degree distributions of network influence the cascading failures are not modeled in the cascading failures model?

A1. Since the word "model" in our previous version might have been misleading, it is replaced to "process" rather than modeling cascading failures. We apologize for any confusion.

Q2. What does the result "In addition, the networks have high efficiency similar to SF networks" in the Abstract specifically mean?

A2. We have removed the insufficient part from the abstract.

Q3. There are several abbreviations that require explanation, for example, BA.

A3. We have added an explanation about Barabási Albert (BA) in section "Continuously changing degree distribution."

For Reviewer #2

Q1. The abstract is too long and contains too many details that should be moved to the introduction or the methods section. Please shorten the abstract to about 150 words and focus on the main motivation, results and implications of your work.

A1. The abstract is shorten in 64 words.

Q2. The introduction lacks a clear statement of the research problem and the research gap that your paper aims to fill. Please provide more background information on the existing literature on network robustness and cascading failures, and explain how your work differs from or extends previous studies.

A2. In order to bridge bring a fresh perspective on the interplay of degree distributions, loops, and tolerance against cascading failures, the guideline is given in the section "Introduction".

Q3. The methods section is not well explained and lacks some important information. For example, how did you generate the networks with different degree distributions? What are the parameters and assumptions of your cascading failure model? How did you measure the efficiency of the networks? Please provide more details and references for your methods and justify your choices.

A3. We added the detailed explanations of how to generate the networks in line 69-73 on the section "Continuously changing degree distributions". In addition, the tolerance parameter α included in Motter model on simulation is described in section "Result", line 174, and for network efficiency calculated by Eq. (4) in the section "Cascading failures".

Q4. The results section is too descriptive and does not provide enough analysis and interpretation of your findings. For example, why do networks with narrower degree distributions have higher tolerance and efficiency than scale-free networks? What are the underlying mechanisms or principles that explain this phenomenon? How do your results compare with other studies on network robustness and cascading failures? Please discuss these aspects more thoroughly and provide some figures or tables to illustrate your results.

A4. An explanation of why networks with narrower degree distributions are more tolerant than scale-free networks is found in lines 186 - 214. Additionally, the efficiency comparison is described in lines 215 -223. To explain the mechanism and principle, some sentences have been added in lines 186 - 223. Furthermore, network efficiency is closely related to the average shortest path length, which is discussed in lines 105-113 on section "Continuously changing degree distributions". Finally, existing studies about cascading failures concentrates on the pin-points of SF networks and ER random graphs. A comparison between them and the networks with narrower degree distributions of this study can be found in lines 186 - 214 as noted earlier.

Q5. Cascading failures is a complex process, and under the same attack, the evolution of cascading failures can have multiple possible paths. How does your research take this into account? And please elaborate on the multiple evolution processes of network cascading failures under three different attacks.

A5. In this study, we are choosing routes deterministically. Considering that there's also the option of randomly choosing from multiple shortest paths, I have added the sentence "Remember that, after removing either AN=1 or 50 nodes as triggers, the flows between and are equally distributed to multiple shortest paths of a same length by Eq. (1) in the determinant choosing. Therefore, stochastic choosing from multiple shortest paths is beyond our scope of this study.", lines 157-159 on the section "Cascade failures".

Q6. In my opinion, for the resilience of complex networks, it is not that the narrower the degree distribution, the better, nor is it the wider the better. Instead, it is about finding its balance point. Your article does not explain this part very clearly. How narrow should the degree distribution be for optimal resilience?

A6. It is suggested that a random regular graph with 0 width has the optimal robustness of connectivity. To clarify the optimal tolerance for cascading failures, perturbation analysis will be required with huge numerical computations as a part of future works. It is added in lines 244-246 on the section "Conclusion".

For Editor

Style

We have ensured the revision met the style requirements. The template is in the following URL. https://journals.plos.org/plosone/s/latex

2. Acknowledgments

According to your suggestion, the Acknowledgments Section has been replaced manuscript. The Funding statement remains unchanged from the current version.

3. Upload Figures

Fig 1, 4, and 5 are uploaded.

We believe that the revisions made have strengthened the overall content and improved the presentation of manuscript. We are confident that the updated version meets the high standards of PLOS ONE and addresses the concerns raised during the review process.

Once again, we would like to express our gratitude for your valuable feedback and guidance. We believe that your suggestions have significantly contributed to enhancing the quality of our work. Please find attached the revised manuscript marked with changes for your convenience.

Thank you for spending your time and consideration.

Sincerely,

Ryota Kusunok

Graduate student

Japan Advanced Institute of Science and Technology

ryotakusunoki@hotmail.com

+8190-7791-9989

---

## [Decision Letter · Decision Letter 1]

7 Nov 2023

PONE-D-23-15565R1Investigating stronger tolerant network against cascading failures in focusing on changing degree distributionsPLOS ONE

Dear Dr. Kusunoki,

Thank you for submitting your manuscript to PLOS ONE. After careful consideration, we feel that it has merit but does not fully meet PLOS ONE’s publication criteria as it currently stands. Therefore, we invite you to submit a revised version of the manuscript that addresses the points raised during the review process.

We look forward to receiving your revised manuscript.

Kind regards,

Qing-Chang Lu

Academic Editor

PLOS ONE

Journal Requirements:

Additional Editor Comments:

There are still minor revisions to be made based on the reviewers' comments.

Reviewers' comments:

Reviewer's Responses to Questions

**Comments to the Author**

1. If the authors have adequately addressed your comments raised in a previous round of review and you feel that this manuscript is now acceptable for publication, you may indicate that here to bypass the “Comments to the Author” section, enter your conflict of interest statement in the “Confidential to Editor” section, and submit your "Accept" recommendation.

Reviewer #1: All comments have been addressed

Reviewer #2: All comments have been addressed

2. Is the manuscript technically sound, and do the data support the conclusions?

Reviewer #1: Yes

Reviewer #2: Yes

3. Has the statistical analysis been performed appropriately and rigorously? 

Reviewer #1: Yes

Reviewer #2: Yes

4. Have the authors made all data underlying the findings in their manuscript fully available?

Reviewer #1: (No Response)

Reviewer #2: Yes

5. Is the manuscript presented in an intelligible fashion and written in standard English?

Reviewer #1: Yes

Reviewer #2: Yes

6. Review Comments to the Author

Reviewer #1: Q1 means that how does link betweenness centralities affect the spread of cascading failures by affecting node load Li(t)？This is suggested to be further explained in the section on Cascading failures. In addition, is the value of Li, gst and gst(i) changed with the spread of cascading failures? Li(0) should also be explained.

Reviewer #2: The article presents an interesting and novel study on the effect of changing degree distributions on the robustness and efficiency of networks against cascading failures. The authors use two models of preferential and inverse preferential attachment to generate networks with different degrees of heterogeneity, and compare them with the classical Erdős–Rényi random graph model. They also consider different types of attacks and measure the damage and performance of the networks after cascading failures.

The article is well-written, clear, and concise. The methods and results are explained in sufficient detail and supported by relevant references. The figures are informative and illustrate the main findings of the study. The discussion and conclusion are coherent and consistent with the results.

The article makes some contribution to the field of network science, as it provides new insights into the role of degree distribution in enhancing the tolerance and efficiency of networks against cascading failures. The article also has practical implications for designing stronger and more resilient network infrastructures in various domains.

I have no major criticisms or suggestions for improvement. I only have some minor comments that the authors may consider to address before publication:

1）In the abstract, it may be better to avoid using abbreviations such as SF and LCC, as they may not be familiar to all readers. Instead, spell out the full terms or use more general descriptions.

2）In the introduction, it may be helpful to briefly explain what cascading failures are and why they are important to study, as some readers may not be familiar with this concept.

3）In the methods section, it may be useful to provide some examples or references for real-world networks that have power-law, exponential, or narrower degree distributions, to motivate the choice of models and parameters.

4）In the results section, it may be interesting to compare the network efficiency before and after cascading failures, to see how much efficiency is lost due to the damage. It may also be worth discussing how the network efficiency relates to other network properties, such as clustering coefficient or average path length.

I agree to accept this research article for publication in PLOS ONE, as I think it meets the journal’s criteria of originality, rigor, relevance, and impact.

7. PLOS authors have the option to publish the peer review history of their article (what does this mean?). If published, this will include your full peer review and any attached files.

Reviewer #1: No

Reviewer #2: No

---

## [Author Response · Author response to Decision Letter 1]

21 Dec 2023

Thank you for taking time to review our manuscript titled "Investigating stronger tolerant network against cascading failures in focusing on changing degree distributions" submitted to PLOS ONE under Manuscript ID PONE-D-23-15565. We greatly appreciate your insightful feedbacks, which have significantly contributed to improving the quality of our work.

We have carefully considered each of your comments and suggestions, and we are pleased to submit the revised version of our manuscript along with this response letter.

We explain the revisions made:

For Reviewer #1

Q1 means that how does link betweenness centralities affect the spread of cascading failures by affecting node load Li(t)？This is suggested to be further explained in the section on Cascading failures. In addition, is the value of Li, gst and gst(i) changed with the spread of cascading failures? Li(0) should also be explained.

For the 1st and 2nd questions, my answers are NO and YES, respectively. Thus, I have added a sentence on lines 137-139 explaining that link betweenness centrality is used to measure the amount of bypass routes. , , are recalculated at each step of the cascading failure simulation, which I have noted on lines 139. Additionally, I have included a description of , the initial load of each node, on lines 141-142.

For Reviewer #2

Q1. In the abstract, it may be better to avoid using abbreviations such as SF and LCC, as they may not be familiar to all readers. Instead, spell out the full terms or use more general descriptions.

I have avoided using abbreviations in the abstract according to the suggestion.

Q2. In the introduction, it may be helpful to briefly explain what cascading failures are and why they are important to study, as some readers may not be familiar with this concept.

I have added the explanation about cascading failures and their significance on lines 50-57.

Q3. In the methods section, it may be useful to provide some examples or references for real-world networks that have power-law, exponential, or narrower degree distributions, to motivate the choice of models and parameters.

I have added real-world examples of power-law distributions on lines 3-7. For exponential and narrower distributions, please refer to lines 38-41. Each of these additions can be found in the Introduction.

Q4. In the results section, it may be interesting to compare the network efficiency before and after cascading failures, to see how much efficiency is lost due to the damage. It may also be worth discussing how the network efficiency relates to other network properties, such as clustering coefficient or average path length.

According to the suggestion, I have added Fig 6 that shows the network efficiency $E$ before and after cascading failures triggered by localized random, max-degree, and max-load node removals in randomized networks after generations by PA and IPA in changing degree distribution. In addition, we mention that the relationship between efficiency and other network properties, such as the clustering coefficient, remains a topic for future works.

---

## [Editor Report · Decision Letter 2]

27 Dec 2023

Investigating stronger tolerant network against cascading failures in focusing on changing degree distributions

PONE-D-23-15565R2

Dear Dr. Kusunoki,

We’re pleased to inform you that your manuscript has been judged scientifically suitable for publication and will be formally accepted for publication once it meets all outstanding technical requirements.

Kind regards,

Qing-Chang Lu

Academic Editor

PLOS ONE

Additional Editor Comments (optional):

Please address the further comments of the reviewers.
---

## [Editor Report · Acceptance letter]

4 Jun 2024

PONE-D-23-15565R2 

PLOS ONE

Dear Dr. Kusunoki, 

I'm pleased to inform you that your manuscript has been deemed suitable for publication in PLOS ONE. Congratulations! Your manuscript is now being handed over to our production team.

Kind regards, 

on behalf of

Dr. Qing-Chang Lu 

Academic Editor

PLOS ONE